# Electric-Field Control of Spin Diffusion Length and Electric-Assisted D’yakonov–Perel’ Mechanism in Ultrathin Heavy Metal and Ferromagnetic Insulator Heterostructure

**DOI:** 10.3390/ma15186368

**Published:** 2022-09-14

**Authors:** Shijie Xu, Bingqian Dai, Houyi Cheng, Lixuan Tai, Lili Lang, Yadong Sun, Zhong Shi, Kang L. Wang, Weisheng Zhao

**Affiliations:** 1Fert Beijing Institute, Ministry of Industry and Information Technology Key Laboratory of Spintronics, School of Integrated Circuit Science and Engineering, Beihang University, Beijing 100191, China; 2Department of Electrical and Computer Engineering, University of California, Los Angeles, CA 90095, USA; 3Shanghai Key Laboratory of Special Artificial Microstructure, Pohl Institute of Solid State Physics, School of Physics Science and Engineering, Tongji University, Shanghai 200092, China; 4Hefei Innovation Research Institute, Anhui High Reliability Chips Engineering Laboratory, Beihang University, Hefei 230013, China

**Keywords:** electric-field control, magnetic materials, ionic liquid gating, ultra-thin heavy metal, spin relaxation time, spintronics

## Abstract

Electric-field control of spin dynamics is significant for spintronic device applications. Thus far, effectively electric-field control of magnetic order, magnetic damping factor and spin–orbit torque (SOT) has been studied in magnetic materials, but the electric field control of spin relaxation still remains unexplored. Here, we use ionic liquid gating to control spin-related property in the ultra-thin (4 nm) heavy metal (HM) platinum (Pt) and ferromagnetic insulator (FMI) yttrium iron garnet (Y_3_Fe_5_O_12_, YIG) heterostructure. It is found that the anomalous Hall effect (AHE), spin relaxation time and spin diffusion length can be effectively controlled by the electric field. The anomalous Hall resistance is almost twice as large as at 0 voltage after applying a small voltage of 5.5 V. The spin relaxation time can vary by more than 50 percent with the electric field, from 41.6 to 64.5 fs. In addition, spin relaxation time at different gate voltage follows the reciprocal law of the electron momentum scattering time, which indicates that the D’yakonov–Perel’ mechanism is dominant in the Pt/YIG system. Furthermore, the spin diffusion length can be effectively controlled by an ionic gate, which can be well explained by voltage-modulated interfacial spin scattering. These results help us to improve the interface spin transport properties in magnetic materials, with great contributions to the exploration of new physical mechanisms and spintronics device.

## 1. Introduction

The major goal of modern spintronic is to process information via manipulation electron spins and charges in solid-state nanoscale devices. The key to realizing spin devices is how to efficiently control the spin current. During recent years, enormous progress has been achieved in this area. For example, effectively controlling spin via an electric field could help to maintain the historic rates of progress that are occurring in computational power [1,2]. An electric control spin current is also useful for low-dissipation logic circuits [3], frequency-tunable microwave nano-oscillators [4,5] and ultralow-power spintronics [2]. In spin current devices composed of HM/FMI heterostructure, the FMI is an ideal material that allows the transport of spin current, but not charge current [6]. The top HM layer serves as the spin injector and converted the spin into current via the inverse spin Hall effect and detected by electrical measurement [6,7]. However, the spin current flow in HM layer has a decay length, which will affect the propagation of the spin.

The ionic liquid Gate voltage [8] is very useful for regulating the interface spin transport property between nonmagnetic metal Pt and ferromagnetic insulator YIG [9,10,11,12]. The ionic liquid composed of N,N-diethy1-N-(2-methoxyethy1)-N-methylammonium (DEME+) and bis (trifluoromethylsulfony1)-imide (TFSI−) have high carrier densities over 1014/cm [2], which can product large interfacial electric fields more than 1 × 106 V/cm [13]. Yttrium iron garnet (Y_3_Fe_5_O_12_, YIG) is an ideal ferromagnetic insulator, due to its high Curie temperature, low damping and long spin-transmission length [6,14]. Strong spin–orbit coupling layer Pt and ferromagnetic insulator have plentiful spin-related physical mechanisms, such as the spin Hall effect [15,16], spin pumping [17], spin Hall magnetoresistance [18] (SMR), magnetic proximity effect [19] (MPE), weak anti-localization [20] (WAL), and topological Hall effect [21] (THE). The spin diffusion length (SDL) in the NM layer, standing for the length scale of traveling spin memorizes, can strongly affect the spin current absorption procession [22]. Therefore, it is necessary to apply gate voltage to control the SDL.

Here, we get the spin scattering time via the weak anti-localization (WAL) effect [23], which comes from quantum diffusion mechanism in disordered conductors with strong spin–orbit interaction [24,25]. The electric field can effectively enhance the spin scattering time, resulting in a larger spin diffusion length (λs). In addition, as the electric-field-modulated electron momentum scattering time (te) increases, the spin scattering time will be suppressed, which indicates that the DP mechanism is dominant. This means that the lack of space inversion symmetry will destroy the spin degeneracy and generate an effective magnetic field, which will lead to interface spin precession at YIG/Pt heterostructure. The present work will help us to understand the electric-field-modulated spin relaxation mechanism and provide an attractive method to control the spin diffusion length, which will be useful for energy-efficient spin devices [26,27,28,29]. 

## 2. Experiments

### 2.1. Materials Preparation

YIG is a synthetic garnet with ferromagnetic properties. The iron ions occupy two octahedral and three tetrahedral sites, resulting in magnetic behavior (Figure 1a). First, 80 nm-thick single-crystalline YIG films were grown on (1 1 1)-oriented single crystalline Gd_3_Ga_5_O_12_ (GGG) substrates via pulsed laser deposition (PLD) using the krypton fluoride coherent excimer laser (λ_laser_ = 248 nm, f = 8 Hz). The GGG substrates were kept at 650 °C for 30 min to remove the surface contamination under the oxygen pressure of 3 Pa. Then, the YIG films were deposited and annealed at 810 °C for 4 h under the oxygen pressure of 6 × 10^4^ Pa, to obtain a high-quality single-crystalline structure (Figure 1b). Then, the sample was naturally cooled to room temperature and transferred to magnetron sputtering to grow 4nm Pt and 2 nm HfO_2_. The oxide layer was mainly used to protect the sample. Pt and HfO_2_ were fabricated via magnetron sputtering at the argon pressure of 0.35 Pa.

### 2.2. Sample Measurement and Ionic Liquid Voltage Control

The samples were patterned into Hall devices, and a drop of ionic liquid was then dripped on the top of the HfO_2_ layer. After the gate voltage was applied, an electric double layer (EDL) accumulating charge carriers was formed on the top of the HfO_2_. The leakage current was below 1 µA at room temperature. The EDL is conserved at the solid state below 210 K. We applied voltage at 300 K and maintained the voltage while cooling down to 5 K. Hereafter, when removing the gate voltage at 5 K, the electric field effect still exists. Therefore, the current leakage should be zero at 5 K. Other physical properties, such as thickness, characterized X-ray reflectivity (XRR), magnetization hysteresis loops, spontaneous magnetization, transverse resistance R_xy_, and longitudinal resistance R_xx_ were measured using a vibrating sample magnetometer (VSM) and a physical property measurement system (Quantum Design PPMS-9T system).

## 3. Results and Discussion

### 3.1. Non-Volatile Electric Field Control

The Pt(4.0 nm)/YIG(80 nm) heterostructure was fabricated into a standard Hall bar to investigate the magneto-resistance and the Hall resistance. The stable EDL was formed on top of the Pt surface, as shown in Figure 1c, by employing the same electric cooling procedure, which can efficiently modulate the interface spin current. Magnetic properties of YIG were measured at room temperature via vibrating sample magnetometer (VSM) of Lake Shore Incorporation. The out-plane M-H loop of the YIG shows that the magnetization is around 150 emu cm^−3,^ as shown in Figure 1d, close to the bulk value [30]. 

Figure 1e shows the sheet resistance modulation, with gate voltage at 5 K. The resistance increases as the gate voltage decreases, and the observed hysteresis can be attributed to slow ion motion and the interfacial trap states [31]. The non-volatile voltage-modulated resistance can be explained by the charge accumulation at the interface between YIG and Pt [32,33]. The gate voltage can not only change the charge accumulation, but also has a large modulation of spin-related properties. After the positive gate voltage is applied, the anomalous Hall effect (AHE) was enhanced. However, for negative voltages, the anomalous Hall effect changes are particularly weak due to non-volatile electric field control property, as shown in the illustration in Figure 1f. The anomalous Hall signal at high *H* can be explained well by the Langevin function [9,12]:(1)$$\;R_{xy}=R_{S}\times L\left(\frac{mH}{k_{B}T}\right)+R_{H}H$$

The first and the second terms correspond to the AHE and the ordinary Hall effect (OHE); *m* is the magnetic moment of Pt atoms; *k_B_* is the Boltzmann constant; *T* is the temperature; and *R_S_* and *R_H_* are the AHE resistance and the ordinary Hall resistance, respectively. The anomalous Hall signal is enhanced because the electric-field can induce spin-polarized Pt [9,12].

### 3.2. Electric-Field Control Spin Relaxtion

To study the potential for using the electric field to control the spin scattering time and the spin diffusion length in the spin-polarized Pt, we studied the weak localization (WL) and weak anti-localization (WAL) related to the quantum diffusion mechanism [23]. The longitudinal resistance was measured for H along the z direction at different temperatures, as shown in Figure 2a. Above 45 K, positive magnetic resistance (MR) showing quadratic H-dependence was observed in a high-H region; this is characteristic of ordinary MR, related to Lorentz force [33]. Below 45 K, negative resistance in a high-H was observed due to WAL. Then, we found that the MR can be effectively regulated by the gate voltage, as shown in Figure 2b. The MR at 6 V has more than doubled compared to the MR at 0 V, indicating that the electric field effectively regulates weak anti-localization effects. 

Figure 2c shows the RT curve at different voltages. This resistance rise is almost proportional to lnT at low temperature, indicating a quantum corrections effect in disordered conductors [19,34,35,36]. Finally, in magnetic fields and under strong spin–orbit interaction, as shown in Figure 2d, the quantum correction to the sheet conductance can be fitted by following the S.Hikami-A. I. Larkin-Y. Nagaoka (HLN) equation [20,23]:(2)$$\Delta \sigma \left(H\right)=\sigma \left(H\right)-\sigma \left(H=0\right)\;=\frac{e^{2}}{2\pi^{2}\hbar }\left[\psi \left(\frac{1}{2}+\frac{B_{1}}{\mu_{0}H}\right)-\frac{3}{2}\psi \left(\frac{1}{2}+\frac{B_{2}}{\mu_{0}\;H}\right)+\frac{1}{2}\psi \left(\frac{1}{2}+\frac{B_{3}}{\mu_{0}\;H}\right)-\ln\left(\frac{B_{1}\sqrt{B_{3}}}{B_{2}^{3/2}}\right)\right]$$
where *B*_1_ = *B*_0_ + *B*_so_; *B*_2_ = 3/4*B*_so_ + *B*_p_ and *B*_3_ = *B*_p_. Here, *B*_0_, *B*_so_, and *B*_p_ (or *B*φ) are effective magnetic fields for elastic, spin–orbit, and phase coherence scattering, respectively. The *B*_p_ mainly characterizes the effective field of electron–phonon interaction and electron-electron interaction in disordered electronic systems; *B*_so_ mainly describes the effective field of spin–orbit coupling. The magnetic resistance at 5 K for YIG/Pt can be fitted very well using Equation (2). The parameter *B*_0_ = 46.5 T for different gate voltage indicated that electric-field control of the elastic scattering effect field is negligible. The spin–orbit-effective field *B*_so_ = 7.5 T at 0 V indicated the spin–orbit scattering caused by the interface magnetic spin exchange interaction. The phase coherence scattering effective field *B*_p_ = 1.495 T at 5 K indicated magnetic scattering at the interface of YIG/Pt. The YIG’s magnetism is in a saturated state under the fitting of a high magnetic field, so the magnetic scattering and phase coherence scattering are very weak. Therefore, the phase coherence scattering effective field is much lower than the spin–orbit scattering effective field.

The electric control of spin–orbit scattering time (t_s_) is shown in Figure 3a. The spin–orbit scattering time t_i_ = h/(4eDB_i_), (I = p, so, e) significantly decreases with the increasing positive gate voltage, although it is nearly constant at negative gate voltage (D is the diffusion coefficient). The t_e_ can be derived using the equation t_e_ = D*m/E_f_. The t_e_ and t_s_ are 2.56 ps and 0.055 ps, respectively, at 5 K, which is comparable to t_e_ and t_s_ in single crystal Pt [17]. In addition, t_e_ and t_s_ can be effectively modulated by the electric field, but exhibits contrary regulation, as shown in a,b, due to DP spin relaxation mechanism, with the frequent momentum scattering events suppressing spin relaxations. 

It is worth noting that R_H_, R_S_, D, B_so_ and B_φ_ are achieved by transverse and longitudinal resistance. The diffusion coefficient D (=v^2^_F_τ_e_ with v_F_ being Fermi velocity) can be obtained by D = −E_F_ ·R_H_ · σ/e, in which E_f_ is the Fermi energy and R_H_ is the ordinary Hall effect. The ordinary Hall effect and anomalous Hall effect can be calculated by Equation (1), as shown in Figure 3c,d. After a gate voltage of 6 V was applied, the OHE coefficient changed significantly from −6.52 (10–11 Ω·m·T^−1^) at 0 V to   −7.72 (10–11 Ω·m·T^−1^), as shown in Figure 4c. Because the sign and magnitude of the OHE coefficient strongly depend on the competition between the s-type electrons near the center point Γ and the d-type holes near the XW5 33, the results in Figure 4c also indicate that the voltage can effectively control the density of states near the Fermi level. The anomalous Hall resistance R_S_, as shown in Figure 3d, increases three times with positive voltage regulation, due to the magnetic moments induced by the electric field [9]. After the analysis of longitudinal resistance by Equation (2), we can achieve the B_so_ and B_φ_ with different V_g_, as shown in Figure 3e,f. The phase coherent effective field and the spin–orbit coupling effective field exhibit the same change trend under voltage modulation. This proved that the EDL’s electric field can control the spin–orbit, along with the change in B_so_.

## 4. Conclusions

### 4.1. Electric-Field Control of Spin Diffusion Length and Electric-Assisted D’yakonov–Perel’ Mechanism in YIG/Pt

Finally, we can calculate the spin–orbit scattering rate Ґ_so_ by using this equation: Ґ_i_=1/t_i_ (I = so, φ). The spin–orbit scattering rate can be effectively controlled by the electric field in YIG/Pt heterostructure, as shown in Figure 4a, which was derived from the effective modulation effect of the EDL’s electric field on the interfacial spin scattering. The spin diffusion length λs = (√3/2)√(D × t_so_), which was based on an almost spherical Fermi surface [37,38], can be effectively controlled by gate voltage, as shown in Figure 4b. Interestingly, a positive voltage leads to an increase in the spin scattering time. Finally, voltages can effectively increase the spin diffusion length, which gives us a deeper understanding the spin scattering procession. To further clarify the spin relaxation mechanism of the YIG/Pt heterostructure, the applied positive electric field promotes electron scattering time, but strongly suppresses spin scattering, which can be explained using the DP model, as shown in Figure 4c. The blue points are the measured t_so_ and te on the Hall bar device. The blue line and red line are the values calculated by the DP and EY model, respectively. The EY mechanism is basically caused by bulk or interface impurity scatterings; more electron momentum scattering chances make the spin relaxation faster. For the DP mechanism induced by spatial inversion symmetry breaking, more electron momentum scattering makes spin relaxation slower, which is in contrast with the EY mechanism [25]. Our measured data can match the DP model very well. The calculated blue line is derived from the DP mechanism spin–orbit scattering equation: t_so_ = (Һ/∆_R_)^2^ (1/t_e_), with Δ_R_ representing the spin-splitting energy created by the interface Rashba SOI. The Δ_R_ of the gate controlling YIG/Pt interface at 5 K is 1.726 × 10^−3^ (eV), which reflects that the interfacial spin scattering between Pt and YIG makes the main contribution to the Rashba spin–orbit interaction (SOI). This work helps us to better understand the spin relaxation mechanism and promote the development of spintronics devices. 

### 4.2. Spin Diffusion Length for Different Metals

The spin diffusion length, which can quantify the decay behavior of the pure spin current in the propagation process, is an important parameter for characterizing the spin Hall effect. For Pt/YIG heterostructure, the measured SDL by WAL effect is 4.1 nm, as shown in Table 1. If we compare the SDL value at 6 V with the same WAL method, we obtain a longer value, λ = 4.1 nm, due to voltage-modulated larger spin scattering time. The spin diffusion length measured by our experiment is basically consistent with the results measured by other groups, as shown in Table 1.

### 4.3. Summary

In conclusion, we applied the gate voltage to change the magnetic-moment-induced anomalous Hall effect. The anomalous Hall effect in high magnetic fields can be well explained by the Langevin function [9,12]. The magnetic moment of platinum ions is believed to stem from the proximity effect, which behaves only in local magnetic order. Thus, the lack of long-range ferromagnetic order leads to super-paramagnetic behavior for the platinum ions, as reported previously [12]. The local magnetic moment of Pt cluster can be treated as a classical macro-spin, with a large local moment (3.5 μB in our work). In addition, the gate voltage can change the Fermi surface via electric-field control of the OHE coefficient. The electron scattering time could become enhanced, because the positive gate voltage shifted upwards the Fermi level and increased the density of state of the s-like band. At the same time, the spin–orbit scattering time will be decreased by the analysis of the HLN equation. The electron scattering and spin scattering can be modulated by voltage and show different trends. These phenomena are mainly dominated by the DP model for 4 nm Pt/YIG derived from Rashba SOI, not the EY mechanism. We use the method of changing the gate voltage as the main means of detecting the spin relaxation mechanism. Finally, the spin diffusion length at the YIG/Pt interface can be effectively enhanced by the gate voltage, which is inseparable from the interfacial spin scattering. This work has great significance for the entrance of effective electric field modulation of spin relaxation in metal, which is also an important step in the design of spintronic devices.

## Figures and Tables

**Figure 1 materials-15-06368-f001:**
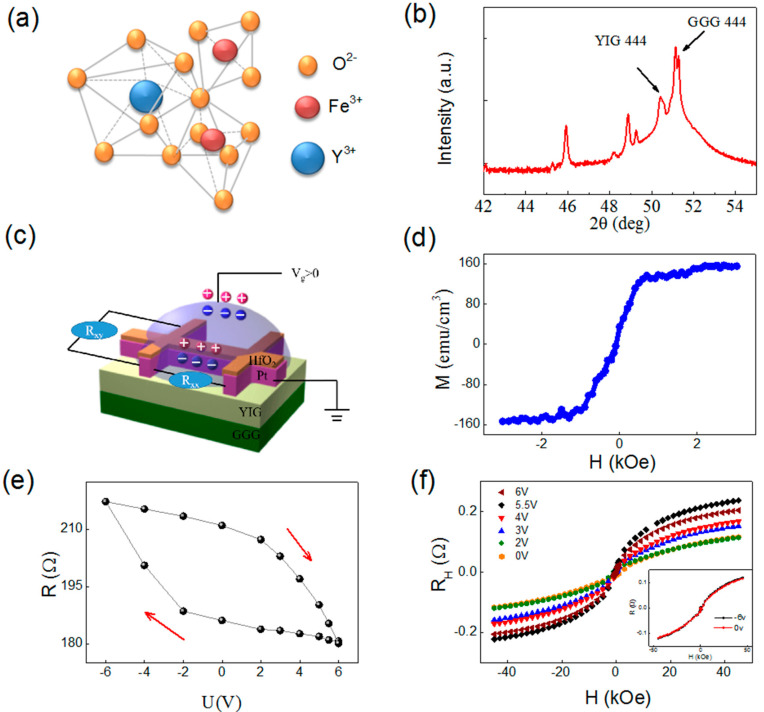
(**a**) Atomic structure of YIG. (**b**) XRD spectra of YIG (80.0 nm)/GGG. (**c**) Schematic picture of the Hall bar measurement combining ionic liquid top gate. (**d**) Magnetization hysteresis loop of YIG/GGG with an in−plane magnetic field. (**e**) The resistance R versus gate voltage at 5 K. (**f**) The Hall resistance versus the external magnetic field at 5 K. The measured data are fitted by Equation (1) after considering the anomalous Hall effect and the ordinary Hall effect.

**Figure 2 materials-15-06368-f002:**
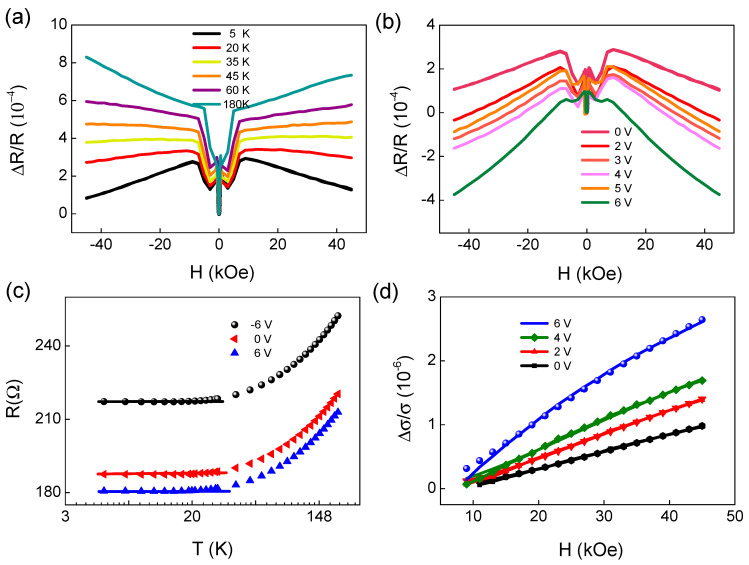
(**a**) Longitudinal magnetic resistance ∆R = R_xx_(H) − R_xx_ (H = 0) versus the external magnetic field H at different temperature with Vg = 0 V. (**b**) Longitudinal magnetic resistance ∆R = R_xx_(H) − R_xx_(H = 0) versus H curves at 5 K with different gate voltage Vg. (**c**) Temperature (T) dependence of sheet resistance (R) for Pt/YIG; the temperature scale is logarithmic. (**d**) H dependence of magneto-conductance ∆σ_sheet_ =1/R (H) − 1/R (H = 0) for Pt/YIG at 5 K. Here, SMR contribution observed in a low-H region is subtracted. The solid line is the fit to weak anti-localization.

**Figure 3 materials-15-06368-f003:**
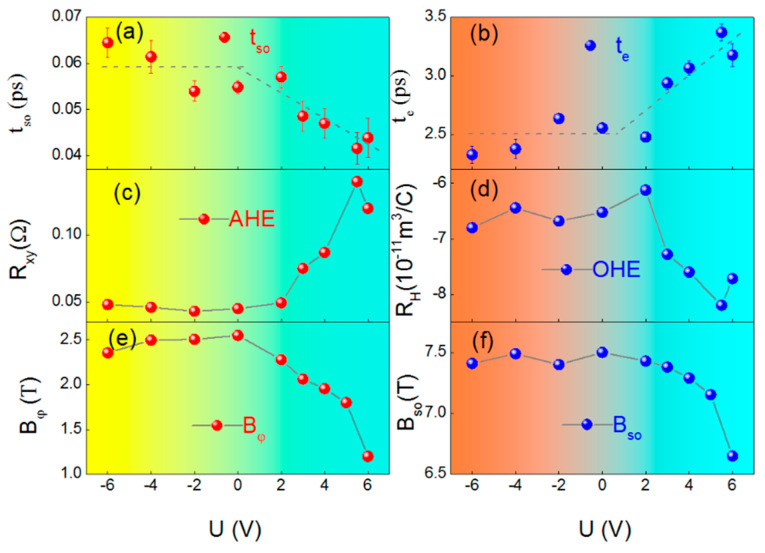
(**a**–**f**) The gate voltage dependencies of the spin–orbit scattering time, electron momentum scattering time, the AHE resistance, the ordinary Hall coefficient, spin–orbit, and phase-coherence-scattering effective field at 5 K.

**Figure 4 materials-15-06368-f004:**
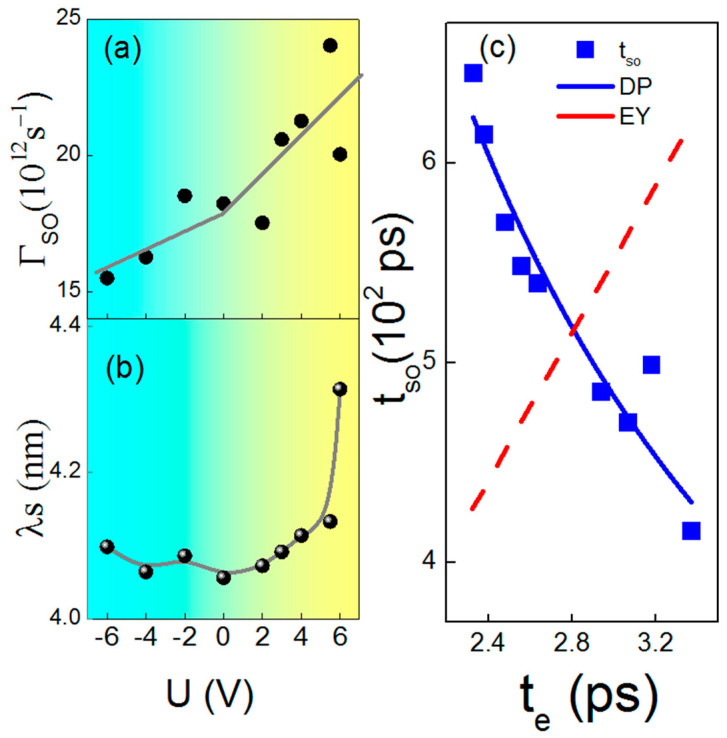
(**a**,**b**) The gate voltage U dependence of spin–orbit scattering frequency and spin diffusion length. (**c**) The spin–orbit scattering time t_so_ as a function of t_e_. The blue line is a reciprocal fit of the data points.

**Table 1 materials-15-06368-t001:** Spin diffusion length for Pt from the literature and this work using different methods.

Materials	T (K)	λ (nm)	Method	Ref
YIG/Pt (0 V)	5 K	4.05	WAL	This work
YIG/Pt (6 V)	5 K	4.31	WAL	This work
FeNi/Cu/Pt	300 K	2	Spin absorption	[39]
FeNi/Cu/Pt	10 K	3.4	Spin absorption	[39]
FeNi/Pt	300 K	1.2	Spin pumping	[40]
FeNi/Pt	8 K	1.6	Spin pumping	[40]
FeNi/Cu/Au	300 K	32	Spin absorption	[39]
FeNi/Cu/Au	10 K	53	Spin absorption	[39]
FeNi/Au	5 K	35	Spin pumping	[40]

## Data Availability

Not applicable.

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
