# Peer review of "Electric-Field Control of Spin Diffusion Length and Electric-Assisted D’yakonov–Perel’ Mechanism in Ultrathin Heavy Metal and Ferromagnetic Insulator Heterostructure"

_materials, 2022, doi:10.3390/ma15186368_

Round 1

Reviewer 1 Report

Electric-Filed Control of Spin Diffusion Length and Electric-Assisted D'yakonov-Perel' Mechanism in Ultrathin Heavy Metal and Ferromagnetic Insulator Heterostructure

The subject of the paper is Electric-Filed Control of Spin Diffusion Length and Electric-Assisted D'yakonov-Perel' Mechanism in Ultrathin Heavy Metal and Ferromagnetic Insulator Heterostructure_SiO2 are the method to control of magnetic order, magnetic damping factor and spin-orbit torque (SOT). The result also usefull for spin transport properties in magnetic materials,However, some recommendations should be taken into account for publication:

Originality/Novelty

While the Electric-Filed Control of Spin Diffusion Length and Electric-Assisted D'yakonov-Perel' Mechanism are novel for this work into heavy metal.

Quality of Presentation

The overall presentation quality is acceptable. The article is, on the whole, well-written; the English used in the paper is understandable.

-          Methodology

the method is not specifically mentioned in this paper, however it can improved to add more detail fabrication process.

-          Results and Discussion

It also advisable to show the fabrication output of real device in order to compare the model provided.

All the equations need to be allign back and also the figures.

Conclusions

The conclusions are straightforward. This needs to be improved. Include more specifics about the outcomes that were achieved.

Overall Merit:

The article certainly has merit. For the rest, I believe that the article is organised in a logical and understandable manner.

Reviewer 2 Report

1- The Abstract is somewhat dry; it should include more actual results from the manuscript.
2- The Authors should also consider rephrasing some of the paragraphs, to improve the manuscript in terms of the English language.
3-
Could the authors compare their results with the properties of the other materials?

For the above comments I will recommend this manuscript as amajor revision.

Reviewer 3 Report

The paper “Electric-Filed Control of Spin Diffusion Length and Electric-Assisted D'yakonov-Perel' Mechanism in Ultrathin Heavy Metal and Ferromagnetic Insulator Heterostructure” by Shijie Xu, Bingqian Dai, Houyi Cheng, Lixuan Tai, LiLi Lang , Yadong Sun, Zhong Shi, Kang L Wang, Weisheng Zhao is devoted to the study of electric field influence on spin relaxation in abovementioned heterostructures using ionic liquid gating. From my point of view the results presented in the work are quit interesting and worth the publication in Materials. But before the publication some issues should be addressed:

  1. No reason to present the hysteresis loop for the magnetic field in the film plane (Fig.1b) because all the following measurements have been done for perpendicular configuration. Better to include the plot with perpendicular hysteresis loop to make a possibility for reader to compare these data with the plots presented in Figs.1 and 2.
  2. Why the Langevin function is used in Eq.1 and what the reason to talk about superparamagnetism here. It just natural to use Brilluen function for platinum ions. Anyway it should be clarified.

The paper is well written accept the Conclusions. I think this part should be reworken.

Round 2

Reviewer 2 Report

Please correct the scale bar of x axis at figure no 1 (b) is that with Oe or KOe , in the first version it's KOe but in the 2nd with Oe so which one is correct

Why the authors didn't presented the results by the XRD for YIG single-crystalline and the morphology structure for the single crystalline YIG  it will be better if these results are presented it will gives more strength to the work 
